# Effect of Ten Weeks of Creatine Monohydrate Plus HMB Supplementation on Athletic Performance Tests in Elite Male Endurance Athletes

**DOI:** 10.3390/nu12010193

**Published:** 2020-01-10

**Authors:** Julen Fernández-Landa, Diego Fernández-Lázaro, Julio Calleja-González, Alberto Caballero-García, Alfredo Córdova Martínez, Patxi León-Guereño, Juan Mielgo-Ayuso

**Affiliations:** 1Laboratory of Human Performance, Department of Physical Education and Sport, Faculty of Education, Sport Section, University of the Basque Country, 01007 Vitoria, Spain; julenfdl@hotmail.com (J.F.-L.); julio.calleja.gonzalez@gmail.com (J.C.-G.); 2Department of Cellular Biology, Histology and Pharmacology, Faculty of Health Sciences, University of Valladolid, Campus de Soria, 42003 Soria, Spain; diego.fernandez.lazaro@uva.es; 3Department of Anatomy and Radiology, Faculty of Health Sciences, University of Valladolid, Campus de Soria, 42003 Soria, Spain; albcab@ah.uva.es; 4Department of Biochemistry, Molecular Biology and Physiology, Faculty of Health Sciences, University of Valladolid, Campus de Soria, 42003 Soria, Spain; a.cordova@bio.uva.es; 5Faculty of Psychology and Education, University of Deusto, Campus of Donostia-San Sebastián, 20012 San Sebastián, Guipúzcoa, Spain; patxi.leon@deusto.es

**Keywords:** muscle recovery, lactate threshold, muscle mass, body composition, aerobic power, sport nutrition, supplementation

## Abstract

Creatine monohydrate (CrM) and β-hydroxy β-methylbutyrate (HMB) are common ergogenic aids in the field of sports and are frequently used in an isolated way. However, there are a few studies that have investigated the effect of combining both supplements on different variables related to performance, with controversial results. Therefore, the main purpose of this study was to determine the efficacy and the degree of potentiation of 10 weeks of CrM plus HMB supplementation on sports performance, which was measured by an incremental test to exhaustion in elite male traditional rowers. In this placebo-controlled, double-blind trial, 10-week study, participants (*n* = 28) were randomized to a placebo group (PLG; *n* = 7), CrM group (0.04 g/kg/day of CrM; *n* = 7), HMB group (3 g/day of HMB; *n* = 7) and CrM-HMB group (0.04 g/kg/day of CrM plus 3 g/day of HMB; *n* = 7). Before and after 10 weeks of different treatments, an incremental test was performed on a rowing ergometer to calculate the power that each rower obtained at the anaerobic threshold (WAT), and at 4 mmol (W4) and 8 mmol (W8) of blood lactate concentration. There were no significant differences in WAT and W4 among groups or in body composition. However, it was observed that the aerobic power achieved at W8 was significantly higher in the CrM-HMB group than in the PLG, CrM and HMB groups (*p* < 0.001; η2*p* = 0.766). Likewise, a synergistic effect of combined supplementation was found for the sum of the two supplements separately at WAT (CrM-HMBG = 403.19% vs. CrMG+HMBG = 337.52%), W4 (CrM-HMBG = 2736.17% vs. CrMG+HMBG = 1705.32%) and W8 (CrM-HMBG = 1293.4% vs. CrMG+HMBG = 877.56%). In summary, CrM plus HMB supplementation over 10 weeks showed a synergistic effect on aerobic power (measured as WAT, W4, and W8) during an incremental test but had no influence muscle mass.

## 1. Introduction

Adequate training and a diet adapted to a specific sporting discipline play a key role in athletes reaching maximum performance [1]. In addition, acceptable supplementation could increase performance by helping athletes to recover from previous efforts [2] or by developing substrates or pathways of specific energy use for the actual sport capacities [3,4]. There are several supplements used to promote muscle recovery [5] through the replacement of energy substrates, such as creatine monohydrate (CrM) [6] or β-hydroxy β-methylbutyrate (HMB) [7,8], among others [9].

CrM is one of the most well-known and studied supplements related to physical performance and health [6]. Some authors have shown improvements in endurance capacity (expressed by the individual lactate threshold) regardless of the effect of intensive training [10], however, the results of studies on aerobic capacity are quite inconsistent [11]. CrM improves aerobic capacity mainly by increasing the creatine-phosphocreatine (Cr-PCr) shuttle, which leads to a higher yield of myocellular ATPases, an increase in PCr re-synthesis, the accumulation of inorganic phosphorus, Ca^2+^, H^+^ and ADP, greater availability of amino acids, inhibition of glycolysis and a possible increase in neuromuscular performance. [12]. Equally, muscle glycogen levels can be positively affected by CrM through the inhibition and/or activation of certain glycogen synthase regulatory proteins, highlighting the IGF-I/Akt-PKB/GSK3 pathway, the possible inhibition of AMPK and cell swelling [13] which are essential in glycolytic sports. Moreover, CrM improves recovery stimulating muscle protein synthesis by the activation of signaling cascades and an increase in the expression of proteins involved in these processes and inactivation and/or reduction in the expression of proteins with ergolytic functions [14], increasing testosterone levels [15] and/or reducing the post-training lactate (LA) concentration [16], lactate dehydrogenase (LDH) [17] and creatine kinase (CK) [17], which are essential to achieve the desired training adaptation and hence, the opportunity to train more.

HMB, a metabolite from leucine, which is one of the three essential branched-chain amino acids, is another supplement the has been widely studied [7,8]. This ergogenic aid improves endurance capacity [18,19,20], enhances mitochondrial biogenesis by activation of gamma co-activator 1-alpha α (PGC-1α), thus promoting higher fat oxidation [21]. Furthermore, HMB may improve muscle glycogen synthesis indirectly by enhancing the insulin effect and amplifying phosphorylation [22]. In addition, HMB may be applied to enhance the muscle mass and strength of skeletal muscles in physically active individuals who exercise [23], increasing muscle protein synthesis [24] by upregulation of mammalian target of rapamycin (mTOR) [7]. Besides, HMB may also reduce blood cortisol levels [25] and decrease the LA concentration [18], CK [26] and LDH [26].

These two supplements have been investigated individually; however, the combination of both is a common strategy in various sports fields. However, to the best of our knowledge, only a few studies have shown the degree of potentiation of this combination, but they have not identified whether their effects are synergistic or additive. For example, some authors have used beta-alanine plus CrM supplementation and have shown a synergistic effect on the physical working capacity at the neuromuscular fatigue threshold [27,28]. Although there are a few studies examining the action produced by the combination of CrM plus HMB, the results obtained are controversial with regard to sport performance [29,30,31,32,33,34]. While some of studies did not show changes in muscular strength, endurance and aerobic and anaerobic ability, others found improvements in muscular strength and in aerobic power [35]. Nevertheless, according to the authors’ knowledge, there are no previous studies that have quantified the degree of potentiation of CrM plus HMB on sport performance.

Thus, we hypothesized that the combination of CrM plus HMB would enhance performance more than CrM or HMB separately. The main purpose of this study was to determinate the efficacy and the degree of potentiation of 10 weeks of supplementation with a mix of 0.04 g/kg/day (≈3 g/day) of CrM plus 3 g/day of HMB on aerobic power measured by an incremental test to exhaustion in elite male traditional rowers.

## 2. Material and Methods

### 2.1. Participants

Twenty-eight elite male traditional rowers (30.43 ± 4.65 years; 23.92 ± 1.85 kg/m^2^ and 8.3 ± 1.15% of fat mass) who belonged to a top rowing club from the First Trainer League in Spain (ACT), with more than 5 years of high level traditional rowing experience participated in this double-blind and placebo-controlled trial. All rowers performed the same team-monitored practice sessions for 6 days/week. The sessions were 1.5 h/day, (including rowing practice, preventive and strength individual training and recovery protocols) and ran for 10 weeks during the rowing season (competitive period with two official rounds of rowing per week). Further, the registered dietitian-nutritionist of the club developed a personalized diet for each rower. The diets were proposed using previously established energy and macronutrient guidelines for adequate athletic performance, and were based on training load, personal characteristics and intolerances of each participant [36].

A medical examination was performed before the study began in order to verify that the participants did not have any disease or previous injury. No participants had any diseases, and none of them smoked, drank alcohol or took medications, which would alter hormone response. Likewise, to avoid the possible interference of other nutritional supplements on the different variables measured in this investigation, a 2-week washout period was introduced. During the investigation period, the athletes only took the assigned supplement and the recovery shake consisting of carbohydrates and proteins.

All of the participants received a physical examination, were fully informed of all aspects of the study, and signed a statement of informed consent. This research was designed in accordance with the Declaration of Helsinki (2008), with the Fortaleza update (2013) [37] and approved by the Human Research Ethics Committee at the Basque Country University, Vitoria, Spain with the number M10/2017/247.

### 2.2. Experimental Protocol and Evaluation Plan

This study was designed as a randomized and placebo-controlled, double-blind trial in order to analyze the effects of 10 weeks of oral supplementation of 0.04 g/kg/day of CrM; 3 g/day HMB; 0.04 g/kg/day of CrM plus 3 g/day HMB or placebo on sports performance measured by an incremental test to exhaustion [38]. The proposed doses were chosen based on the safety and efficacy of creatine and HMB supplementation in exercise, sport, and medicine [6,7].

The participants were randomly assigned by SPSS software to four different groups using a stratified block design. An independent statistician generated the randomization sequence: (1) Placebo group (PLG; *n* = 7; height: 184.9 ± 2.4 cm and body mass: 81.9 ± 6.3 kg), (2) Group treated with 0.04 g/kg/day of CrM (CrMG: *n* = 7; height: 183.4 ± 7.8 cm and body mass: 81.2 ± 5.0 kg), (3) Group treated with 3 g/day of HMB (HMBG; *n* = 7; height: 185.5 ± 10.1 cm and body mass: 79.9 ± 12.2 kg) and (4) Group treated with 0.04 g/kg/day of CrM plus 3 g/day of HMB (CrM-HMBG; *n* = 7; height: 181.6 ± 4.3 cm and body mass: 78.0 ± 4.7 kg). All participants attended the laboratory (at 8:30 a.m.) for blood collection at two specific points during the study: (1) at baseline (T1), and (2) post-treatment (T2—the day after 10 weeks of treatment).

The four groups took supplementation or placebo during the 6 days of weekly training together with a chocolate recovery shake (1 g/kg of CHO + 0.3 g/kg protein) in the half hour after finishing the exercise [39]. No substance was added to the PL, but the athletes were unaware of this situation. On the off days, the rowers took the same dose of supplements 30 min before going to bed. None of the participants used any pre-workout substances. An independent nutritionist from outside the club made the shakes with the individual supplementation, so each rower and researcher did not know which supplementation was being taken. Moreover, each day the same nutritionist verified that all rowers had complied with the protocol for taking the supplements. The CrM was obtained from Creapure^®^ powder, while the HMB was obtained from HMB-Ca FullGas^®^ (Fullgas Sport, S.L, 20115 Astigarraga, Guipúzcoa (Spain)).

### 2.3. Incremental Power Tests

To evaluate the athletes’ performance, an incremental test was carried out at T1 and T2. The two test sessions were carried out at 6:30 p.m. in a covered sports hall with standard conditions (temperature: 21 °C and humidity: 60%) to keep the constants equal in both tests. The tests were performed after a standardized 15-min warm-up. The warm-up included 10 min of constant rowing with two 1-min accelerations (at 3 min and 5 min of the warm-up) and 5 min of accelerations and injury prevention drills consisting of general movements, dynamic/static stretching and core stability. All rowers consumed 3 g/kg of CHO 1–4 h before both tests [36].

The incremental test [38] was performed on an indoor rowing ergometer (Concept II system, Model D, Morrisville, VT, USA), on which the seat was fixed to remain static during the test [40]. The test was performed with stages of 3 min of progressive intensity until fatigue with rest intervals of 30 s between stages in order to obtain samples of LA in the lobe of the ear. The initial workload was 100 W. Once the test started and at each stage, the rower was asked to maintain the constant intensity (W) and constant strokes [41]. The intensity was increased by 40 W in each subsequent stage until exhaustion. Exhaustion was defined as the rower’s inability to sustain three consecutive strokes at the stipulated power. All tests were valid maximum stress tests using the standard criteria for rowers [42].

### 2.4. Blood Lactate Concentrations

The LA samples were obtained by samples (5 μL) from the earlobe of each rower before beginning the test and at the end of each 3-min stage. The LA was determined by a Lactate Scout analyzer (EKF Diagnostics^®^, Penarth, Cardiff, UK) following the manufacturer’s instructions [43]. To avoid inter-analyzer variability, the same analyzer was used for both tests in all participants. The validity of the analyzer was guaranteed by verifying the measured values with the lactate standards according to the manufacturer’s instructions [43].

### 2.5. Determination of Thresholds

After obtaining the different LA values at each stage of the incremental protocol, they were represented graphically as a continuous function against time. Then, the power that each rower achieved at the anaerobic threshold (WAT), at 4 mmol (W4) and 8 mmol (W8) was extrapolated. For the WAT calculation, the D-max method was used [44,45].

### 2.6. Anthropometry

Anthropometric measurements were taken following the protocol of the International Society for the Advancement of Kinanthropometry (ISAK) [46]. Additionally, the same internationally certified anthropometrist (ISAK level 3) took the measurements for all participants. All measurements were taken in duplicate to establish within-day retest reliability. If the difference between the duplicate measures exceeded 5% for an individual skinfold, a third measurement was taken. The mean of duplicate or the median of triplicate anthropometric measurements were used for all analyses. Height (cm) was measured using a SECA^®^ measuring rod, with a precision of 1 mm, while BM (kg) was assessed by a SECA^®^ model scale, with a precision of 0.1 kg. Body mass index (BMI) was calculated using the equation BM/height^2^ (kg/m^2^). Six skinfolds (mm) from the triceps, subscapular, suprailiac, abdominal, front thigh and medial calf were measured with a Harpenden^®^ skinfold caliber with a precision of 0.2 mm, and the sum of these was calculated. The girth (cm) of the relaxed arm, mid-thigh and calf were measured with a narrow, metallic and inextensible Lufkin^®^ model W606PM measuring tape with a precision of 1 mm. Fat mass (FM) was calculated using the Carter equation [47] and the muscle mass (MM) was calculated by the Lee equation [48].

### 2.7. Dietary Assessment

All participants were informed about proper food tracking by the same trained nutritionist-dieticians. They instructed the athletes on two validated methods of dietary recall [49]. The first method was to complete a food frequency questionnaire (FFQ) at T2, which has been previously utilized for sport populations [50]. This FFQ, which asked the participants to recall their average intake based on certain “frequency” categories over the previous 10 weeks, included 139 different foods and drinks, arranged by food type and meal pattern. Frequency categories were based on the number of times an item was consumed per day, per week or per month. Daily consumption of energy (kcal) and each macronutrient in grams was determined by dividing the reported intake by the frequency in days.

The second method was a 7-day dietary recall at T1 and T2 of the 7 days before the test, which was used to examine whether the results of this recall were similar to that of the FFQ. If the participants had weighed food, then that data was used for the recall; however, if the weighing of food was not possible, serving sizes consumed were estimated from the standard weight of food items or by determining portion size by looking at a book with 500 photographs of foods. Food values were then converted into intake of total energy, macronutrients and micronutrients by a validated software package (Easy diet^©^, online version 2019). This software package was developed by the Spanish Centre for Higher Studies in Nutrition and Dietetics (CESNID), which is based on Spanish tables of food composition [51]. Likewise, total energy and macronutrients intake in relation to each kg of BM was calculated for each athlete.

### 2.8. Statistical Analysis

All variables are presented as the mean ± SD. The percentage change between the T1 and post-treatment T2 tests of the variables was calculated as Δ (%): ((T2 − T1)/T1 × 100) for each study group.

The Shapiro–Wilk test was used to determine the normality of the data (*n* < 50) for all continuous variables, therefore, we used parametric formulas. Besides, Levene’s test was applied to measure the homoscedasticity of the variances. Mean levels of Δ (%), dietary intake and power output from incremental tests at T1 and T2 were compared across supplementation consumption categories using one-way analysis of covariance with the supplementation category as the fixed factor. A Bonferroni post-hoc test was applied for pairwise comparisons among groups. Likewise, differences from T1 to T2 in each group were assessed by a parametric dependent *t*-test. Moreover, a two-way repeated measure of analysis of variance (ANOVA) test was used to examine interaction effects (time × supplementation group) among the supplementation groups (PLG, CrMG, HMBG and CrM-HMBG) for power output by incremental testing.

Effect size among participants were calculated using a partial eta square (η2*p*). Since this measure is likely to overestimate the effect size, the values were interpreted according to Ferguson [52], which indicates no effect if 0 ≤ η2*p* < 0.05, a minimum effect if 0.05 ≤ η2*p* < 0.26, a moderate effect if 0.26 ≤ η2*p* < 0.64, and a strong effect if η2*p* ≥ 0.64.

The following calculation was used to express the variables in the CrMG, HMBG and CrM-HMBG as a percentage change from the PLG condition [53].

Normalized change (%) = (Treatment (CrMG, HMBG or CrM-HMBG)/Control (PLG) − 1) × 100.

Using the additive model, stressor (in fact all variable) interactions are categorized as either synergistic or antagonistic. Significant interactions suggest the effect size of one variable has been reduced (antagonistic) or accentuated (synergistic) by the presence (or effect) of the other whereas additive effects are shown during net stressor independence, i.e., no interaction [53]. Interactions are best illustrated using variables A and B: (1) Additive: A and B combined = A + B individually; (2) Synergistic: A and B combined > A + B individually; (3) Antagonistic: A and B combined < A + B individually; (4) Nullifying: A and B combined = A or B individually; (5) Multiplicative: A and B combined = A × B individually.

The analyses were performed using SPSS software version 24.0 (SPSS, Inc., Chicago, IL, USA) and Microsoft Excel (Microsoft Excel Software version 19). Statistical significance was indicated when *p* < 0.05.

## 3. Results

During the study, the athletes did not show significant statistical differences (*p* > 0.05) in energy and macronutrient intake among groups (Table 1). The energy intake was approximately 45 kcal/kg in each study group. In the same way, the intake of proteins, fats and CHO was ≈1.9 g/kg; 1.5 g/kg and 6.0 g/kg, respectively, in each study group.

Table 2 displays the anthropometry and body composition data at both T1 and T2 in each of the study groups. There were no significant differences in the group-by-time in body mass, the sum of six skinfolds, fat mass (kg) and muscle mass (kg) (*p* > 0.05). Regarding body mass and fat mass, a significant decrement was found in all groups during the study (*p* < 0.05). However, a significant decrement was found in muscle mass in the PLG between the two study moments (T1: 33.3 ± 4.3 vs. T2: 32.7 ± 4.1 kg; *p* < 0.05; η2*p* = 0.160).

Table 3 shows the values for the power obtained at different intensities during the incremental test at both T1 and T2. Significant differences (*p* < 0.05) can be seen in the group-by-time for W8 (*p* < 0.001; η2*p* = 0.766). However, there were no significant differences in the group-by-time for WAT and W4.

In addition, significant increases (*p* < 0.05) between study points were observed for WAT in the HMBG (T1: 238 ± 40 vs. T2: 253 ± 35 W; η2*p* = 0.168) and the CrM-HMBG (T1: 238 ± 22.73 vs. T2: 264 ± 19 W; η2*p* = 0.168), for W4 in CrM-HMBG (T1: 236 ± 29 vs. T2: 263 ± 19 W; η2*p* = 0.181) and for W8 in the CrMG (T1: 300 ± 20 vs. T2: 314 ± 21 W; η2*p* = 0.766), HMBG (T1: 295 ± 45 vs. T2: 314 ± 49 W; η2*p* = 0.766) and CrM-HMBG (T1: 288 ± 21 vs. T2: 331 ± 35 W; η2*p* = 0.766).

Despite there being no group differences in terms of the mean for absolute power output at T1 and T2, the ANOVA did detect significant interactions regarding percentage change (Figure 1). Specifically, there were significantly greater increases in favor of the CrMG (+5 ± 2%); HMBG (+6 ± 3%) and CrM-HMBG (+15 ± 5%) compared to the PLG (−1 ± 3%) at T1 to T2 for W8. In addition, there were significantly greater increases in favor of the CrM-HMBG compared to the CrMG and HMBG (*p* < 0.01).

The results in Table 4 showed the changes in WAT, W4, and W8 in the all groups supplemented with respect to PLG, after 10 weeks. A synergistic effect of combined supplementation was found for the sum of the two supplements on WAT (CrM-HMBG = 403% vs. CrMG+HMBG = 338%), W4 (CrM-HMBG = 2736% vs. CrMG+HMBG = 1705%) and W8 (CrM-HMBG = 1293% vs. CrMG+HMBG = 878%) using the equation: Synergistic = A and B combined >A + B individually [51].

Also, when comparing the synergism found in the combination CrM-HMBG, it was observed (Table 4) that the most synergistic effect (CrM-HMBG–(CrM+HMBG)) was found on W4.

## 4. Discussion

To the authors’ knowledge, there are only a few studies that examine the combined supplementation of CrM plus HMB [29,30,31,32,33,34]. Five of these were carried out on intermittent team sports (three in rugby [30,31,34], one in basketball [32] and one in soccer [33]), sports that are characterized by the combination of short, high intensity actions with low intensity actions. The results regarding athletic performance are controversial [29,31,32,33,34]. Two of the studies did not find significant differences in aerobic performance (multistage aerobic capacity test) [34], anaerobic performance (60 s maximal anaerobic capacity test) [34], muscular strength (3RM test) [31], muscular endurance (maximum number of chin-ups to exhaustion) [31], peak power (10-s leg power test) [31] and total work (10-s leg power test) [31]. However, three of them showed improvements in peak power and mean power (running anaerobic speed test) [33], in the accumulative strength tests (1-RM) [29] and in the relative maximal and total anaerobic power (triple Wingate test) [32]. These results cannot be compared with those obtained in this study, because this study is the only one that measures aerobic performance in an incremental test (the traditional rowing incremental test), which is a good predictor of aerobic performance [54]. Moreover, although they are not clear, some authors have indicated that an increase in muscle mass is a benefit of both these supplements [55,56]. However, the present study indicates that supplementation with CrM plus HMB did not show differences in the body composition (muscle mass and fat mass) of athletes. These results might be explained by the high training level of the participants and the strict nutritional control during the study.

The LA is produced in muscle cells during exercise when glucose is oxidized as a process of anaerobic glycolysis [57]. When the intensity of exercise is increased, LA is associated with the impossibility of continuing exercise [57]. Interestingly, the LA decrease when the same intensities are performed, enhances endurance capacity [57]. The supplementation of CrM plus HMB for 10 weeks showed that at the same LA level (8 mmol/L), a significantly greater work power was realized when supplements were taken individually. In the same vein, Zajac et al. [32] and Faramarzi et al. [33] observed a significant increase in relative maximal and total aerobic power as measured by a triple Wingate test, and in peak power measured by a running anaerobic speed test (RAST). However, in contrast with these studies, the most important finding of our research was the synergistic effect of combined supplementation found in all of the performance tests (WAT, W4, W8). Therefore, the two supplements could use different effect pathways, and as a result, improve performance (expressed as power indices).

Besides, CrM ingestion increased muscle total creatine, and therefore influences the Cr-PCr shuttle, which may lower the LA reduction by lowering glycolysis (saving glycogen) at the same intensity, improving anaerobic capacity [6]. This process consists of resynthesizing phosphocreatine (PCr) though free Cr in mitochondria, which results in a higher energy availability during exercise. Free Cr reacts with the mitochondrial isoenzyme of the creatine kinase (mi-CK), burning an adenosine triphosphate (ATP), in the intermembrane space of mitochondria. The resynthesizing processes PCr, which is used by the myofibril to produce ATP after the reaction with the muscular isoenzyme of the creatine kinase (MM-CK). The ATP used to create the PCr is transformed into ADP that crosses the mitochondrial intermembrane through adenine nucleotide translocase (ANT) to enter into the mitochondrial inner membrane-matrix space, where it reacts with the ATP synthase consuming protons (H^+^), reproducing ATP. That metabolite passes through ANT to the intermembrane space to react with another Cr molecule, starting the cycle again [12]. HMB can increase the gene expression of peroxisome proliferator-activated receptor PGC-1α, changing the fiber transformation type (driving fast-to-slow fiber switch) and improving mitochondrial biogenesis, and hence oxidative function to enhance the aerobic capacity [21]. This process consists of augmenting the density and quantity of the muscle cell mitochondria, angiogenesis, and hence increases fat oxidation, which enhances aerobic capacity [58]. These adaptations could save glycogen, which is a limiting factor during endurance exercise, increasing oxidative capacity and reducing LA production by glycolysis [57].

One of the long-term adaptations to endurance training is the capacity to lower the LA levels [59]. Adaptations are promoted after an adequate recovery period, and there are strategies to accelerate the recovery. These supplements could influence recovery by increasing protein synthesis, decreasing protein degradation, muscle glycogen, and improving membrane repair [6]. In particular, protein synthesis may be augmented by HMB and CrM. HMB can increase this parameter through the stimulation of mTOR phosphorylation by some downstream targets such as p70S6k, eIF4E and eIF2B [60], and by increasing the growth hormone–insulin-like growth factor-1 (GH-IGF-1) axis [61]. Protein degradation can be lowered by HMB, decreasing the catalytic activity of the proteasome [62] and also by increasing the GH–IGF–1 axis [61]. Glycogen storage may be increased by both supplements. HMB can enhance glycogen synthesis [22], probably by accelerating the tricarboxylic acid cycle to provide a carbon skeleton for glycogen synthesis. CrM can also enhance muscle glycogen storage by increasing AMP-activated protein kinase (AMPK) phosphorylation, and hence GLUT4 translocation [13]. Sarcolemma can be repaired by HMB and it is converted in HMG-CoA for cholesterol synthesis, and hence it lowers blood muscular damage marker levels, such as CK and LDH [63,64]. CrM can also augment protein synthesis, enhancing the myogenic transcription factor MRF-4 [65] and accordingly, differentiating the satellite cells into myonuclei [66].

### 4.1. Limitations, Strengths and Future Research

The study has limitations, including the small sample size per group (*n* = 7), in total 28 participants; however, it is very difficult to obtain larger samples in elite sports. Moreover, the study was carried out in a controlled environment (randomized, double-blind, nutrition, training), which could be considered a strength of this study. In addition, the diet ingested by the athletes was controlled throughout the intervention process, so that these parameters did not influence the final results and the effects of CrM plus HMB.

For future research, there is a need for more studies on the combination of CrM plus HMB supplementation in endurance-based ~20-min all-out exercise in order to have stronger evidence of the effectiveness of this mix of ergogenic aids on anaerobic lactic performance. Another potential method could be the analysis of the effect of these supplements in women athletes.

### 4.2. Practical Application

The knowledge gained from this study could have practical application for athletes and practitioners who are interested in improving their endurance capacity, given that the intake over 10 weeks of a combination of HMB (3 g/day) and CrM (0.04 g/kg/day) could improve sport endurance capacity.

In the context of post-exercise recovery, this would result in the improvement of anaerobic performance, i.e., both supplements could have the same effect through different mechanisms of action, which fully justifies their combined use. However, our results are mixed (the improvements in WAT, W4, and W8 were not the same) and are strongly influenced by the state of participants’ training, sample size and the specific type of protocol or exercise measurement used. Therefore, more studies are needed to determine the overall efficacy of HMB-CrM supplementation as an ergogenic aid, given the controversy surrounding the studies investigating the effect of HMB-CrM supplementation on anaerobic response-induced muscular performance.

## 5. Conclusions

In summary, oral supplementation with a combination of 0.04 g/kg/day (≈3g/day) of CrM plus 3 g/day of HMB over 10 weeks of training showed a synergistic effect on aerobic power (measured as WAT, W4, and W8) during an incremental test (related to individual lactate threshold). Although both supplements showed a possible improvement in the incremental test separately, a synergic effect was shown when CrM plus HMB are mixed, likely due to their different physiological mechanisms.

## Figures and Tables

**Figure 1 nutrients-12-00193-f001:**
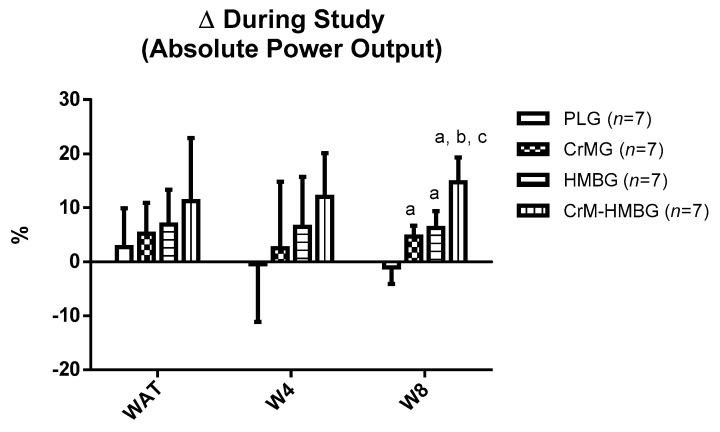
Percentage changes during study on absolute power output of the anaerobic threshold (WAT), 4 mmol/L (W4) and 8 mmol/L (W8) in the four study groups. Data are expressed as means ± standard error. PLG: Placebo group; CrMG: Creatine monohydrate supplemented group; HMBG: HMB supplemented group; CrM-HMBG: Creatine monohydrate plus HMB supplemented group. Δ: ((T2 − T1)/T1) × 100; differences among groups in each test by ANOVA test (*p* < 0.05): a: regarding PLG; b: regarding CrMG; c: regarding HMBG.

**Table 1 nutrients-12-00193-t001:** Energy and macronutrient intake in the four study groups during 10 weeks of study.

	PLG	CrMG	HMBG	CrM-HMBG
Energy (kcal)	3340 ± 350	3358 ± 358	3290 ± 410	3375 ± 395
Energy (kcal/kg)	44 ± 6	45 ± 6	45 ± 6	45 ± 7
Protein (g)	143 ± 24	145 ± 26	141 ± 29	143 ± 26
Protein (%)	17 ± 3	18 ± 3	18 ± 3	17 ± 3
Protein (g/kg)	2 ± 0	2 ± 1	2 ± 1	2 ± 0
Animal protein (g)	84 ± 23	86 ± 25	81 ± 19	86 ± 25
Vegetal protein (g)	60 ± 11	58 ± 15	62 ± 19	59 ± 17
Fat (g)	101 ± 20	103 ± 21	99 ± 21	101 ± 22
Fat (%)	27 ± 4	28 ± 4	27 ± 5	28 ± 5
Fat (g/kg)	2 ± 0	2 ± 1	2 ± 1	2 ± 1
Total carbohydrates (g)	450 ± 55	460 ± 60	459 ± 58	453 ± 61
Carbohydrates (%)	54 ± 5	55 ± 5	55 ± 6	54 ± 5
Carbohydrates (g/kg)	6 ± 1	6 ± 1	6 ± 1	6 ± 1
Fe (mg)	24 ± 7	24 ± 7	24 ± 7	23 ± 8

Data are expressed as mean ± standard deviation. PLG: Placebo group; CrMG: Creatine monohydrate supplemented group; HMBG: HMB supplemented group; CrM-HMBG: Creatine monohydrate plus HMB supplemented group.

**Table 2 nutrients-12-00193-t002:** Anthropometry and body composition data in the four study groups at the baseline (T1) and after 10 weeks (T2).

Group	T1	T2	P (TxG)	η2*p*
Body mass (Kg)				
PLG	81.9 ± 6.3	80.0 ± 5.3 *	0.883	0.028
CrMG	81.2 ± 5.0	78.6 ± 5.4 *		
HMBG	79.9 ± 12.2	77.6 ± 11.1 *		
CrM-HMBG	78.0 ± 4.7	75.5 ± 4.5 *		
6 Skinfolds (mm)				
PLG	51.6 ± 18.9	48.8 ± 16.3	0.790	0.050
CrMG	57.0 ± 6.5	54.7 ± 14.1		
HMBG	54.2 ± 11.4	52.0 ± 13.0		
CrM-HMBG	50.4 ± 7.1	47.4 ± 4.9		
Fat mass (kg)				
PLG	7.3 ± 2.7	6.4 ± 2.3 *	0.207	0.255
CrMG	6.1 ± 0.7	5.8 ± 0.7 *		
HMBG	6.8 ± 1.3	6.3 ± 1.1 *		
CrM-HMBG	6.4 ± 0.8	6.2 ± 0.4 *		
Muscle mass (kg)				
PLG	33.3 ± 4.3	32.7 ± 4.1 *	0.442	0.160
CrMG	31.5 ± 1.9	31.2 ± 2.3		
HMBG	32.8 ± 1.5	32.2 ± 1.1		
CrM-HMBG	34.6 ± 1.3	34.6 ± 1.1		

Data are expressed as mean ± standard deviation. P (TxG): group-by-time interaction (*p* < 0.05). Two-factor repeated-measures ANOVA. *: Significant difference between study points (T1 vs. T2). *p* < 0.05. PLG: Placebo group; CrMG: Creatine monohydrate supplemented group; HMBG: HMB supplemented group; CrM-HMBG: Creatine monohydrate plus HMB supplemented group.

**Table 3 nutrients-12-00193-t003:** Power output at the anaerobic threshold (WAT), 4 (W4) and 8 mmol (W8) in the four study groups at the baseline (T1) and after 10 weeks (T2).

Group	T1	T2	P (TxG)	η2*p*
WAT (W)				
PLG	254 ± 34	259 ± 20	0.228	0.168
CrMG	242 ± 21	253 ± 12		
HMBG	238 ± 40	253 ± 35 *		
CrM-HMBG	238 ± 22	264 ± 19 *		
W4 (W)				
PLG	242 ± 16	241 ± 31	0.196	0.181
CrMG	243 ± 20	247 ± 21		
HMBG	238 ± 45	251 ± 36		
CrM-HMBG	236 ± 29	262 ± 19 *		
W8 (W)				
PLG	317 ± 19	314 ± 24	<0.001	0.766
CrMG	300 ± 20	314 ± 21 *		
HMBG	295 ± 45	31 ± 48 *		
CrM-HMBG	288 ± 21	331 ± 35 *		

Data are expressed as mean ± standard deviation. P (TxG): group-by-time interaction (*p* < 0.05. All such occurrences). Two-factor repeated-measures ANOVA. *: Significantly different between study points (T1 vs. T2) *p* < 0.05. PLG: Placebo group; CrMG: Creatine monohydrate supplemented group; HMBG: HMB supplemented group; CrM-HMBG: Creatine monohydrate plus HMB supplemented group.

**Table 4 nutrients-12-00193-t004:** Determining the effect of the combination of supplements.

Group	CrMG (A)	HMBG (B)	A + B	CrM-HMBG	CrM-HMBG − (A + B)
**WAT**	134%	203%	338%	403%	66%
**W4**	397%	1309%	1705%	2736%	1031%
**W8**	364%	514%	878%	1293%	364%

Data are expressed with respect to change to the placebo group (%) = (Treatment group/placebo (PLG) − 1) × 100 [51]. A + B = Sum of the effects of CrMG (A) and HMBG (B) when participants are supplemented independently.

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
