# Peer review of "Effect of Ten Weeks of Creatine Monohydrate Plus HMB Supplementation on Athletic Performance Tests in Elite Male Endurance Athletes"

_nutrients, 2020, doi:10.3390/nu12010193_

Round 1

Reviewer 1 Report

I would encourage you to have someone proof read  and edit the paper to address structural sentence errors and redundancy in the paper. Why was a crossover design not employed to increase findings? Introduction: Address the mechanism behind Cr in the pathway as you did HMB. Page 2 Line 66...sentence starts "In addition" line 68..next sentence starts "In addition". (Correct these types of redundancies throughout the paper. Page 3 L 112 you state dosages were selected based on effectiveness and safety. What levels of intake for each supplement presents adverse effects? State in paper. Assure all graphs may be understood and interpreted as stand alone items. Were training outcomes controlled during the study? Were subjects permitted to take other supplements during this study period? Address washout periods for subjects coming into the study on other supplements? 

Author Response

Dear Reviewer,

We really appreciate the time you devoted to reading our manuscript and helping us to craft an improved version. We are pleased to clarify your concern which we believe will improve the impact and quality of your work. Please find below our response to your observation. We have made a concerted attempt to systematically address the specific concerns raised for this revision and we have highlighted the alterations to this revision within the manuscript in yellow for your convenience.

In advanced,

King Regards

REVIEWER 1

REVIEWER: I would encourage you to have someone proof read and edit the paper to address structural sentence errors and redundancy in the paper.

AUTHORS: Thanks so much for your help. The manuscript has undergone English language editing by MDPI. The text has been checked for correct use of grammar and common technical terms, and edited to a level suitable for reporting research in a scholarly journal.

REVIEWER: Why was a crossover design not employed to increase findings?

AUTHORS: Thanks for your interest. The main idea of the article was to understand better the effect of long-term supplementation (10 weeks) of creatine + HMB on performance, since the results observed from past research were controversial, when this combination was supplemented from 6 days to 6 weeks. Due to the experimental procedure (4 intervention groups), having performed a crossover trial would has taken longer period than the duration of a sports season. In addition, we have chosen this method due to this study has been carried out during competitive period. In consequence, the resting hormone status oscillation in the different training periods, could produce alterations in the final results. On the other hand, double-blind and placebo-controlled trial are an experimental strategies used to ensure impartiality and avoid errors derived from bias.

REVIEWER: Introduction: Address the mechanism behind Cr in the pathway as you did HMB.

AUTHORS: Thank you for your observation. The mechanism behind Cr by the pathway has been included in the introduction. We have included the next sentence: “The main pathway where CrM could improve aerobic capacity would be increasing the creatine-phosphocreatine (Cr-PCr) shuttle which leads to a higher yield of myocellular ATPases, an increase in PCr resynthesis, in the accumulation of inorganic phosphorus, Ca2+, H+ and ADP, greater availability of amino acids, inhibition of glycolysis and a possible increase in neuromuscular performance. [12]. Equally, muscle glycogen levels could be positively affected by CrM through the inhibition and / or activation of certain glycogen synthase regulatory proteins, highlighting the IGF-I / Akt-PKB / GSK3 pathway, the possible inhibition of AMPK and cell swelling [13] being essential in glycolytic sports. Moreover, CrM could improve recovery stimulating muscle protein synthesis by the activation of signaling cascades and the increase in the expression of proteins involved in these processes and inactivation and / or reduction in the expression of proteins with ergolytic functions [14], increasing testosterone levels [15] and/or the reducing post training LA concentration [16], lactate dehydrogenase (LDH) [17] and creatine kinase (CK) [17], that are essential to achieve the training adaptation and hence the possibility to train more.“

REVIEWER: Page 2 Line 66...sentence starts "In addition" line 68..next sentence starts "In addition". (Correct these types of redundancies throughout the paper.

AUTHORS: Thank you for your observation. The authors have changed the second “in addition” for Besides. Moreover, the authors have corrected these types of redundancies throughout the paper.

REVIEWER: Page 3 L 112 you state dosages were selected based on effectiveness and safety. What levels of intake for each supplement presents adverse effects? State in paper.

AUTHORS: Thank you for your interest. For the best of the authors' knowledge, there is no consensus about the levels of intake that produce adverse effects of each supplement. However, several initial studies about creatine supplementation provided 15–25 g/day of creatine monohydrate for 4 – 12 weeks in athletes engaged in heavy training, without reported side effects [6]. In the same way, studies have found no potential adverse side effects when supplementing with HMB in humans consuming 3–6 grams daily [7,8]. Based on these statements, indicated in both stand positions {6,7], the authors have included the next sentence in Experimental protocol and evaluation plan section: “The proposed doses were chosen based on the safety and efficacy of creatine and HMB supplementation in exercise, sport, and medicine [6,7]”

Kreider, R.B.; Kalman, D.S.; Antonio, J.; Ziegenfuss, T.N.; Wildman, R.; Collins, R.; Candow, D.G.; Kleiner, S.M.; Almada, A.L.; Lopez, H.L. International Society of Sports Nutrition position stand: safety and efficacy of creatine supplementation in exercise, sport, and medicine. Int. Soc. Sports Nutr. 2017, 14, 18. Wilson, J.M.; Fitschen, P.J.; Campbell, B.; Wilson, G.J.; Zanchi, N.; Taylor, L.; Wilborn, C.; Kalman, D.S.; Stout, J.R.; Hoffman, J.R.; et al. International Society of Sports Nutrition Position Stand: beta-hydroxy-beta-methylbutyrate (HMB). Int. Soc. Sports Nutr. 2013, 10, 6. Wilson, G.J.; Wilson, J.M.; Manninen, A.H. Effects of beta-hydroxy-beta-methylbutyrate (HMB) on exercise performance and body composition across varying levels of age, sex, and training experience: A review. Nutr. Metab. 2008, 5, 1–17.

REWIEWER: Assure all graphs may be understood and interpreted as stand alone items.

AUTHORS: Thank you for your observation. In order to assure all graphs may be understood and interpreted as stand-alone items the graphic has been redone.

REVIEWER: Were training outcomes controlled during the study?

AUTHORS: Thank you for your interest. According to the team's schedule, the study outcomes were only analyzed at the control points used in the study. From these data, the strength and conditioning coach individualized the intensity range of the exercise based on this data.

REVIEWER: Were subjects permitted to take other supplements during this study period?

AUTHORS: Thank you for your interest and for this nice detail. During the investigation period the athletes only took the assigned supplement and the recovery shake consisting of carbohydrates and proteins. This sentence has been added in participants section.

REVIEWER: Address washout periods for subjects coming into the study on other supplements? 

AUTHORS: Thank you for your interest. The washout period was 10 days. In this sense, we have added the next sentence in Participant section: “Likewise, to avoid the possible interference of other nutritional supplements on the different variables measured in this investigation, a 10-day washout period was introduced.”

Reviewer 2 Report

This study examined creatine monohydrate and/or HMB on elite rowers for 10 weeks and appropriately demonstrated that creatine m and HMB have a synergistic effect on aerobic power.  As stated in the manuscript, this is a novel take on creatine m and HMB as it thoroughly investigates additive versus synergistic effects on endurance athletes.

The introduction was thorough.  My only comment for the intro would be to cite the "among others" in line 49 and "results of studies on aerobic capacity are quite inconsistent" in line 53. 

The methods are described well and the study design is appropriate for the purpose.  Inclusion of an RD for dietary assessment and then an outside RD for drink mixtures demonstrates the thoroughness of this study.

The results are well laid out in the tables and figure, and discussed appropriately in the results section.

The discussion was thorough and addressed the findings as they relate to the current research.

One point that needs to be addressed (unless I missed it) is, were the participants previously on any of the supplements?  If so, what was the wash-out period before starting the study?

Lastly, the manuscript needs to be carefully edited.  Many sentences were not complete or included extra words or spelling/grammar errors.  For instance, Line 48 states "trough the replace of energy substrates" but I believe it should state "through the replacement of energy substrates". 

Author Response

Dear Reviewer,

We really appreciate the time you devoted to reading our manuscript and helping us to craft an improved version. We are pleased to clarify your concern which we believe will improve the impact and quality of your work. Please find below our response to your observation. We have made a concerted attempt to systematically address the specific concerns raised for this revision and we have highlighted the alterations to this revision within the manuscript in yellow for your convenience.

In Advanced

King Regards

REVIEWER 2

This study examined the intake of creatine monohydrate and/or HMB on elite rowers for 10 weeks and appropriately demonstrated that creatine monohydrate and HMB have a synergistic effect on aerobic power.  As stated in the manuscript, this is a novel take on creatine monohydrate and HMB as it thoroughly investigates additive versus synergistic effects on endurance athletes.

REVIEWER: The introduction was thorough. My only comment for the intro would be to cite the "among others" in line 49 and "results of studies on aerobic capacity are quite inconsistent" in line 53.

AUTHORS: Thank you for your commentary. Good detail, we have added scientific references in  both cases, based on your indications.

REVIEWER: The methods are described well and the study design is appropriate for the purpose.  Inclusion of an RD for dietary assessment and then an outside RD for drink mixtures demonstrates the thoroughness of this study.

AUTHORS: Thank you so much. We really appreciate your comment and feedback.

REVIEWER: The results are well laid out in the tables and figure, and discussed appropriately in the results section.

AUTHORS: Thanks again. We appreciate your comment.

REVIEWER: The discussion was thorough and addressed the findings as they relate to the current research.

AUTHORS: Thank you so much. We appreciate your comment being positive.

REVIEWER: One point that needs to be addressed (unless I missed it) is, were the participants previously on any of the supplements?  If so, what was the wash-out period before starting the study?.

AUTHOR: Thank you for your interest. The washout period was 10 days. In this sense, we have added the next sentence in participants section: “Likewise, to avoid the possible interference of other nutritional supplements on the different variables measured in this investigation a 10-day washout period was introduced.”

REVIEWER: Lastly, the manuscript needs to be carefully edited. Many sentences were not complete or included extra words or spelling/grammar errors. For instance, Line 48 states "trough the replace of energy substrates" but I believe it should state "through the replacement of energy substrates".

AUTHORS: Thanks so much for your help. The manuscript has undergone English language editing by MDPI. The text has been checked for correct use of grammar and common technical terms, and edited to a level suitable for reporting research in a scholarly journal.
